# Exploration of Brain Connectivity during Human Inhibitory Control Using Inter-Trial Coherence

**DOI:** 10.3390/s20061722

**Published:** 2020-03-19

**Authors:** Rupesh Kumar Chikara, Wei-Cheng Lo, Li-Wei Ko

**Affiliations:** 1Department of Biological Science and Technology, College of Biological Science and Technology, National Chiao Tung University, Hsinchu 300, Taiwan; rupesh.bt01g@g2.nctu.edu.tw; 2Center For Intelligent Drug Systems and Smart Bio-devices (IDS2B), National Chiao Tung University, Hsinchu 300, Taiwan; 3Institute of Bioinformatics and Systems Biology, National Chiao Tung University, Hsinchu 300, Taiwan; 4The Drug Development and Value Creation Research Center, Kaohsiung Medical University, Kaohsiung 80708, Taiwan

**Keywords:** electroencephalography (EEG), inhibitory control, brain connectivity, inter-trial coherence (ITC), frontal lobe, temporal lobe, delta band, theta band

## Abstract

Inhibitory control is a cognitive process that inhibits a response. It is used in everyday activities, such as driving a motorcycle, driving a car and playing a game. The effect of this process can be compared to the red traffic light in the real world. In this study, we investigated brain connectivity under human inhibitory control using the phase lag index and inter-trial coherence (ITC). The human brain connectivity gives a more accurate representation of the functional neural network. Results of electroencephalography (EEG), the data sets were generated from twelve healthy subjects during left and right hand inhibitions using the auditory stop-signal task, showed that the inter-trial coherence in delta (1–4 Hz) and theta (4–7 Hz) band powers increased over the frontal and temporal lobe of the brain. These EEG delta and theta band activities neural markers have been related to human inhibition in the frontal lobe. In addition, inter-trial coherence in the delta-theta and alpha (8–12 Hz) band powers increased at the occipital lobe through visual stimulation. Moreover, the highest brain connectivity was observed under inhibitory control in the frontal lobe between F3-F4 channels compared to temporal and occipital lobes. The greater EEG coherence and phase lag index in the frontal lobe is associated with the human response inhibition. These findings revealed new insights to understand the neural network of brain connectivity and underlying mechanisms during human response inhibition.

## 1. Introduction

The ability to suppress an ongoing motor action is known as inhibitory control or response inhibition. It is necessary for the control of executive function of the brain. Usually, inhibitory control is investigated using the stop-signal task or go/no-go task in the laboratory scale. In this study, we utilized auditory stop-signal task to examine the visual and auditory sensory pathways under the inhibitory control of different hands. Moreover, visual and auditory stimuli are used in real environment [1] and we investigated the neural activities of visual and auditory stimuli in the brain during inhibition. In the real-world, we receive many inputs simultaneously from several sensory systems, like visual and auditory stimuli. The process of auditory stimulation is generated in the temporal lobe of the central nervous system, specifically in the primary auditory cortex [2] and the process of visual stimulation occurred in the occipital lobe of the central nervous system, particularly in the primary visual cortex [3]. Therefore, we observed the brain connectivity between the frontal, temporal and occipital lobes of the brain. Previous research reported that coherence has been used to investigate connectivity between two regions of the brain under multisensory integration [4,5]. In our study, we measured brain connectivity in visual and auditory stimuli under human inhibitory control of the left hand as well as the right hand using the phase lag index and the inter-trial coherence (ITC) methods.

Furthermore, recent studies reported that significant electroencephalography (EEG) oscillations of visual perception were found in the alpha (8–12 Hz) and theta (4–7 Hz) band powers [6,7]. The human brain oscillation has been observed under multisensory integration using a decomposition of EEG signals over time and frequency. In resting state, the brain’s EEG signals have been interrupted by visual and auditory stimuli (i.e., multisensory integration effect in the human brain) [8]. In addition, EEG phase synchronization between two brain regions oscillations was measured by calculating the phase relationships between two EEG signals [9,10]. The phase lag index provides direct information of the brain connectivity under multisensory integration, which is not possible for other EEG signal analyses methods, such as event-related potential (ERP) and event-related spectral perturbation (ERSP) [8]. However, previous studies have reported that the phase lag index of brain oscillations is influenced in visual and auditory perception in response to different sensory stimulations [11,12]. In addition, brain dynamics of visual stimulus has been found in the auditory cortex of the brain [13,14,15].

Most previous studies of human response inhibition investigated the EEG neural activities under right-handed subjects with stop-signal task [16,17]. In the current study, we compared the EEG neural activities of both left-hand and right-hand response inhibitions. However, it is not clear that both groups have shown similar or different patterns of EEG oscillation in brain. It was reported that left-handed subjects showed less hemispheric asymmetries than right-handed subjects [18,19]. Studies of electroencephalography (EEG) and functional magnetic resonance imaging (fMRI) have been reported that the frontal lobe was activated during human response inhibition [20,21,22]. In addition, inability of human inhibitory control has been associated with some neurological disorders including attention deficit hyperactivity disorder (ADHD) or obsessive-compulsive disorder (OCD) [23,24]. In this paper, we investigated the cortical neural network under multisensory integration with hand response inhibitions. For the first time, we examined the brain connective by phase lag index and inter-trial coherence methods. We hypothesized that multisensory integration will affect brain connectivity in the frontal, temporal and occipital lobes of the brain.

## 2. Materials and Methods

### 2.1. Participants

Twelve male subjects and one female subject aged from 25 to 30 participated in the stop signal task. A total of 13 subjects joined this study. To eliminate gender differences in the EEG signal analysis [25], all subjects were men in this study. All subjects performed the right hand response (RHR) and left hand response (LHR) inhibitions. In go trials (75%), all subjects were instructed to respond to a square (LHR) and a circle (RHR). In stop trials (25%), an auditory stop signal (beep sound, 750 Hz, 100 ms long) was used to instruct subjects to inhibit both their left hand and right hand responses. All healthy participants were right-handed without hearing, vision impairment or psychological disorders. This study was carried out in accordance with the recommendations of the Institutional Review Board (IRB) of the National Taiwan University, Taipei, Taiwan. The study was approved by the Research Ethics Committee of the National Taiwan University, Taipei, Taiwan. All participants gave their written informed consent in accordance with the Research Ethics Committee of National Taiwan University, Taipei, Taiwan.

### 2.2. Experimental Scenario

The stop-signal task is a race between the go and stop trials. In the stop signal task, frequent go stimuli (75%) and infrequent stop stimuli (25%) were presented, giving the participants realistic feeling of response inhibition, as shown in Figure 1. In this study, the subjects performed an auditory stop signal task, in which the go and stop trials occurred at random. An auditory beep tone signal indicates subjects to inhibit their response in stop trial [20,21]. Moreover, the go-trial was a shape-judgment task, in which subjects need to discriminate between a square and a circle symbol, as shown in Figure 1. In go-trials (75% of trials), only visual stimuli were presented and subjects were instructed to respond to a square and a circle as quickly and accurately as possible. In stop trials (25% of trials), the visual stimuli were followed by an auditory stop signal (beep sound, 750 Hz, 100 ms long) and the subjects were instructed to inhibit their left and right hand responses. The fixation cross sign and visual-auditory stimuli were shown in the center of the computer screen, on a black background. Each go trial was started with a white central fixation cross for 250 ms, followed by a square or a circle symbol for 1000 ms. In the 25% of randomized stop-trials, subjects received a binaural auditory tone over the headphones. This short beep was presented for 100 ms as a stop signal, which triggered the subjects to inhibit their response, regardless of the symbol presented. In the stop trial, a stop signal was presented after a variable of stop signal delay (SSD). The SSD was originally set at 250 ms and it was adjusted continuously by the staircase tracking method. According to the staircase tracking process, when the subject response inhibition was successful, SSD increased by 50 ms; when the subject response inhibition was failed, the SSD decreased by 50 ms. This staircase tracking method increased the complexity of the stop signal task. The objective of the staircase follow-up procedure was to converge on an SSD where subjects successfully inhibited 50% of the stop trials. This allows the calculation of the stop signal reaction time (SSRT) by subtracting the SSD latency from the reaction time (RT) in go trials. However, the presentation of visual and auditory stimuli in the stop signal task began when the participants pressed the enter key on the keyboard. The response keys were used “*Z*” for square (left hand response), and “/” for circle (right hand response). The experimental was terminated when the ESC key was pressed. In this work, 180 go trails and 60 stop trials were executed for each participant.

### 2.3. Acquisition of EEG Signals

The EEG signals were collected from all subjects using a Scan NuAmps Express system (Compumedics USA Inc., Charlotte, NC, USA). The 32-channel EEG cap was used with Ag-AgCl electrodes. The arrangement of all EEG channels were fixed according to the international standard 10-20 system, as shown in Figure 2. To reduce the size of the data and eliminate noise, the EEG signals were sampled at 500 Hz and filtered with a bandpass (1–40 Hz) using an infinite impulse response (IIR) filter before further analysis. The EEG signals were preprocessed using custom routines in MATLAB R2012b (The MathWorks Inc., Natick, MA, USA). Independent component analysis (ICA) is an effective method for removing various types of artifacts, such as eye-movement and eye blinking artifacts, muscle artifacts and environmental noise. To extract clear EEG signals of the human response inhibition, ICA decomposition is a preferred computational method for the separation of the blind source in EEG signals [26,27]. The ICA was performed using the functions of the EEGLAB toolbox (10.2.2.4bVersion, UC San Diego, Swartz Center for Computational Neuroscience, La Jolla, CA, USA) [28]. Each epoch was extracted from 500 to 1300 ms in successful go (SG) and successful stop (SS) trials. Figure 3 shows all steps of EEG signal processing to measure brain connectivity under human inhibition.

### 2.4. Inter-Trial Coherence (ITC) Analysis of EEG Signals

In this study, we used inter-trial coherence method to quantify the degree of phase consistency between two trials of EEG signal, as described in Equation (1), where “*N*” denotes the number of trials; “*t*” represents the time of EEG signal trials, “*f*” stands for the frequency of EEG signal trials and the “*φ*” was a value between –*π* and *π* in EEG signals [28].
(1)$${\mathrm{ITC}}_{\left(t,f\right)}=\frac{1}{N}\overset{k}{\underset{n=1}{\sum }}e^{i\left(\varphi \;k_{\left(t,f\right)}\;2\pi \right)}$$

This method measured the time-frequency analysis trial-by-trial to examine the brain dynamics of inhibition. The results obtained with the ITC method were more accurate than traditional time domain and frequency domain analysis, such as the event-related potential (ERP) and event-related spectral perturbation (ERSP). The ITC method observed the synchronization of EEG signals at a specific latency and frequency from where EEG signals were time locked. In other words, the ITC method decomposed the ERP into their respective phase-locked frequency bands and provided additional information about the relationship between ERP and brain oscillations [8]. In this work, ITC analysis was performed using the ITC functions of the EEGLAB toolbox 10.2.2.4b [28]. The ITC analysis addressed the phase synchronization and desynchronization of EEG signals under response inhibition with visual and auditory stimuli. The ITC value varies from 0 to 1, where a zero indicates the absence of synchronization between two EEG signals while the value 1 represents a perfect synchronization between two EEG signals [6,29].

### 2.5. Brain Connectivity with Phase Locking Value (PLV)

Phase lag index has been recognized as a promising technique to the analysis of brain connectivity. The phase lock value (PLV) is a statistical method that has been used to investigate synchronization or desynchronization between EEG signals, and was used to measure the phase connectivity between the two EEG channels in this work. The Equation (2) we used for the analysis of phase lag index is shown as follows:(2)$$\mathrm{PLV}=\frac{1}{N}\overset{N}{\underset{k=1}{\sum }}e^{j}\left|\varphi 1\left(k\right)-\varphi 2\left(k\right)\right|$$
where “*N*” is the number of samples in the time window considered. The “*φ*1” and “*φ*2” are the phase values of the two EEG signals from two channels. The PLV is always a value between 0 and 1 that reflects how two EEG signals synchronize or synchronize with each other. The phase coherence was calculated using the Hilbert transformation [30]. The phase coherence was observed between eleven pairs of EEG channels including F3-F4, F3-T7, T7-O1, O1-O2, T8-O2, F4-8, F4- F4-O2, F3-O2, F3-O1 and T7-T8 in the frequency range 1–40 Hz. To investigate brain connectivity under response inhibition, the selected 11 pairs of EEG channels covered the whole area of the brain. Most human inhibition related studies have observed that event related with synchronization occurred in theta (4–7 Hz), alpha (8–12 Hz), and beta (13–30 Hz) frequency bands under response inhibition function [31,32,33,34].

## 3. Results

### 3.1. EEG-Inter-Trial Coherence (ITC) Results

Results of the average ITC of all male subjects are shown in Figure 4, Figure 5, Figure 6 and Figure 7, in which non-significant ITC values are colored green while significant values (p < 0.05) are in yellow or red. In each of these inter-trial coherence (ITC) plots, the y-axis is the frequency range (Hz) and the x-axis indicates the time in milliseconds. The ITC plots were acquired for successful go (SG) (visual stimuli), successful stop (SS) (auditory stimuli) trials during left- and right-hand response inhibitions. Besides, to the left side of each ITC plot is a panel with a blue signal that shows the average power of ITC at each frequency. The green and black dotted lines in the left panel show the threshold of ITC significance in each frequency band relative to the baseline period (p < 0.05).

Brain activities related to human inhibitory control were investigated under left- and right-hand response inhibitions in the frontal cortex (F3, FZ, F4). Figure 4 shows the increase of ITC activities in the delta (1–4 Hz) and theta (4–7 Hz) frequency bands, which were observed after visual and auditory stimuli during successful stop (SS) trials in the frontal cortex. These findings are similar to the previous study of inhibitory control [31]. Consequently, the frontal area of the brain is actively associated with the process of inhibitory control. In addition, Figure 4 displays the increase of ITC activities of alpha (8–12 Hz) and beta (13–30 Hz) bands after visual stimuli in successful go (SG) trials. These increases of ITC activities in alpha and beta bands are related to the movement of the hand. The functional role of the frontal cortex in human inhibition show a significant increase of ITC in delta and theta bands under LHR and RHR inhibitory controls as elicited through comparing SS-SG.

Figure 5 presents the increase of ITC activities in the delta (1–4 Hz), theta (4–7 Hz) and alpha (8–12 Hz) bands in the occipital cortex (O1, O2, OZ) with auditory stimuli in successful stop (SS) trials of left and right hand response inhibitions. Figure 5 also shows the increase of ITC activities of the beta band (13–30 Hz) in the occipital cortex after visual perception during successful go (SG) trials of left and right response. Additionally, Figure 6 and Figure 7 display the increase of ITC activities of the delta (1–4 Hz) and theta (4–7 Hz) bands in the right temporal cortex (T8, TP8) with auditory stimuli during successful stop (SS) condition. Figure 6 also revealed increased ITC activities in the delta (1–4 Hz), theta (4–7 Hz) and alpha (8–12 Hz) bands in the right temporal cortex after visual stimulation under the successful go (SG) condition, suggesting a novel neural signature of human response inhibition in the auditory cortex under SS and SG conditions. In addition, increased ITC activities in the delta (1–4 Hz), theta (4–7 Hz) and alpha (8–12 Hz) bands in the left temporal (T7, TP7) cortex under the SS and SG conditions are observed as well (see Figure 7).

### 3.2. Differences in EEG Activities between Male And Female Subjects

Differences in EEG activity between male and female subjects were investigated to show gender differences in the EEG signal analysis. Figure 8 shows the increase of ITC activities in the delta (1–4 Hz), theta (4–7 Hz), alpha (8–12 Hz) and beta (13–30 Hz) frequency bands, which were observed after visual and auditory stimuli during SS-SG condition in the frontal and occipital cortices. Figure 9 displays the increase of ITC activities in the delta (1–4 Hz), theta (4–7 Hz), alpha (8–12 Hz) and beta (13–30 Hz) frequency bands which were detected after visual and auditory stimuli during SS-SG condition in the left temporal and right temporal cortices. The EEG activity patterns in a single female subject were observed to be slightly higher than in male subjects. The EEG activity patterns in a single female subject were observed to be gradually higher than in male subjects. A previous study reported that gender differences in the EEG signal analysis under cognitive activity, they women indicated significantly higher alpha relative power than men during all cognitive conditions, while men revealed significantly higher beta relative power than women [25]. Our study observed similar higher EEG activity patterns of the alpha power band in women than in men.

### 3.3. Brain Connectivity under Human Inhibitory Control using Visual and Auditory Stimuli

The brain dynamics of visual and auditory stimuli under human inhibition involves all regions of the brain. A previous study of time-frequency analysis reported that human inhibitory control is related to the frontal cortex and the pre-supplementary motor area [31]. Therefore, in this study, we developed a brain connectivity model to examine the neural connection between the frontal, central, temporal, parietal and occipital cortices. We examined the phase lock value (PLV) of eleven pairs of EEG channels, including the frontal cortex (F3-F4), left frontotemporal (F3-T7), left temporal-occipital (T7-O1), occipital (O1-O2), right temporal-occipital (T8-O2), right frontotemporal (F4-T8), right frontal-left occipital (F4-O1), right frontal-occipital (F4-O2), left frontal-right occipital (F3-O2), left frontal-occipital (F3-O1) and temporal (T7-T8) cortices. These eleven pairs of EEG channels were selected to measure the brain connectivity under human response inhibition, as shown in Figure 10.

Figure 10A shows the brain connectivity of resting-state in pre-stimulus condition under right-hand response (RHR) inhibition at successful stop (SS) condition. Figure 10B displays the increase of neural connectivity in the frontal cortex (F3-F4), right frontal-left occipital lobes (F4-O1), right frontal-occipital (F4-O2) lobes and occipital (O1-O2) lobe under visual stimuli (SSD) at successful stop (SS) condition. Figure 10C shows the increase of brain connectivity in frontal cortex (F3-F4) electrodes, right frontal-occipital lobes (F3-O1) electrodes, left frontal- right occipital lobes (F3-O2) electrodes, right frontal-temporal lobes (F3-T7) electrodes, right temporal-occipital lobes (T7-O1) electrodes and temporal lobe (T7-T8) electrodes during auditory stimulus (SSRT) at successful stop (SS) condition.

Likewise, Figure 10D shows the brain connectivity of resting-state in pre-stimulus condition under left-hand response (RHR) inhibition at successful stop (SS) condition. Figure 10E displays the increase of brain connectivity in the frontal lobe (F3-F4), the left frontal-right occipital lobes (F3-O2), the right frontal-temporal lobes (F4-T8), the right frontal-occipital lobes (F4-O2) and right temporal-occipital lobes (T8-O2) under the visual stimuli (SSD) at successful stop (SS) condition. Figure 10F shows increase of neural connectivity in the frontal lobe (F3-F4), the left frontal-occipital lobes (F3-O1), the left frontal-left occipital lobes (F3-O2), the left temporal-occipital lobes (T7-O1), the right temporal occipital lobes (T8-O2) and the temporal lobe (T7-T8) during the auditory stimuli (SSRT) at successful stop (SS) condition. However, in this study, all subjects performed both left- and right-hand response inhibition randomly. Therefore, we assumed that all subjects had mixed neural connectivity of LHR and RHR inhibitions. The brain connectivity in the frontal regions we observed agreed with the results of previous studies on human inhibition [35,36].

Moreover, Figure 11A, C shows the brain connectivity of resting-state in pre-stimulus state under RHR and LHR inhibitions in the successful go (SG) condition. Figure 11B presents the increased brain connectivity in the right frontal-occipital (F4-O2) and left temporal-occipital (T7-O1) lobes after visual stimuli (RT). Figure 11D displays the nonsignificant difference in the frontal, temporal and occipital regions of the brain.

## 4. Discussion

In the present study, the neural network of human inhibitory control was deciphered using both visual and auditory perceptions. We examined the brain connectivity between the frontal, temporal and occipital cortices under hand response inhibitions using visual and auditory stimuli. A key finding from the EEG results was the increase of ITC in delta, theta bands, and brain connectivity in frontal-temporal lobes of the brain under the successful stop (SS) condition. In addition, an increased ITC activity in the beta band was observed during the successful go (SG) condition in the frontal lobe. The increased ITC activity of the delta and theta bands in the temporal lobe suggests a novel neural marker for human inhibitory control.

### 4.1. Neural Oscillations under Inhibition with Visual and Auditory Stimuli

The multimodal neural network can be generated by two or more sensory stimulations. For example, when we hear someone’s voice (auditory), we also see their lips moving (visual) [37]. In the real world, simulations to someone do not occur one by one but normally come at the same time, just like watching while listening. In this situation, the EEG dynamics of the brain is very difficult to recognize, such as the discrimination between neuronal activities related to visual from auditory stimuli [38,39,40,41,42,43,44,45]. This study revealed how multisensory information integrates into the human brain under inhibition and the results should be helpful for understanding how auditory inputs affect visual perception and behavior. 

Previous studies reported that visual and auditory sensory inputs can modulate the neural activity in the temporal lobe [46,47,48,49]. In addition, the neurons of the superior colliculus brain had been observed to receive multisensory inputs [50]. Visual and auditory perceptions commonly work together to facilitate the identification and location of sensory input sources in the environment. The superior colliculus brain plays an important role in the relationships between auditory and visual stimuli [1]. Previous study examined the effect of multisensory integration in the theta frequency band at the superior colliculus brain [51,52]. In our work, we found a similar theta ITC increase in the frontal lobe and temporal lobe. The spatial distribution of visual and auditory stimuli in the frontal, central and occipital lobe was studied [53]. According to that work, it was discovered that the delta and theta bands increase with audiovisual stimulation in somatosensory cortex, at least in monkey.

### 4.2. Brain Dynamics under Human Inhibitory Control 

It has been observed that there is increased activities of theta and alpha power bands in frontal and pre-supplementary area under human response inhibition [31]. In the current work, we also discovered similar increased EEG activities of theta and alpha power bands in the frontal cortex of the brain [31]. In addition, we established a brain connectivity neural network model during left- and right-hand response inhibitions. EEG-coherence is a mathematical technique that can be used to determine if two or more regions of the brain have similar oscillatory neuronal activities. Since the 1960s, EEG phase coherence has been applied to evaluate the similarity of the frequency band across EEG signals. In the recent neurological disorders study, it has been used to measure the brain connectivity [54,55].

EEG phase coherence is an analysis of the inconsistency of time alterations between EEG channels. Moreover, the Fourier transform provides a direct association between the time and frequency domains and characterizes the time alteration as a phase difference or phase angle. If the phase angle is stable and constant over time (i.e., phase locked), then the coherence is 1.0, and if the time differences between two time series differ from moment to moment, then coherence will be 0. The EEG phase coherence is often understood as a “connection” analysis, and as a measure of the functional association between two regions of the brain [56,57]. The coherence between EEG channels represents the neural network of brain dynamics under human inhibition. The EEG phase differences are frequently used to calculate “directed coherence,” which is a quantity of the directional flow of information between two EEG electrode sites [58,59]. EEG phase differences were also used to evaluate conduction velocities and synaptic integration times as one increases the inter-electrode distance in different directions [60,61,62,63]. In our work, we investigated the greater EEG phase coherence in the frontal cortex (F3-F4) under human inhibitory control. These findings suggest that all subjects have strong brain connectivity neural network correlated with inhibitions of the left and right hand responses. In addition, we examined the EEG coherence in the temporal and occipital lobes of the brain. Accordingly, inter-trial coherence (ITC) was found maximal in inhibition of right-hand response compared to inhibition of left-hand response. In addition, we observed after auditory stimuli that all subjects showed strong brain modulation compared to visual stimuli. Thus, we established a brain connectivity neural network model under inhibition. In this model, we found some neural network pathways are similar to the previous studies of brain network [35,36]. 

It is noticeable that there were some limitations of the current work. All participants were males and; therefore, it may not be suitable to directly apply the conclusion to female subjects. In the future, we will carry out studies on female subjects and compare the results of brain connectivity of the male versus female groups. The experiment design in our study was adopted from a well-known stop signal task, which used only 2D image and might not be demonstrative enough in actual environments. Our future study may construct a real-environment based experiment scenario in virtual reality (VR) and augmented reality (AR). Before applying these advanced technologies, we could at present conclude that the brain connectivity neural network model under human inhibition with visual and auditory sensory established in this work is sound under normal experimental scenarios, although more researches shall be conducted to fully dissect the underlying significance of the developed model.

## 5. Conclusions

In present study, for the first time, we utilized inter-trial coherence and phase lag index to measure the brain connectivity under human inhibitory control. We have established an inhibitory neural network model. In addition, neural signatures related to human inhibitory control were examined in the frontal cortex and temporal lobe of the brain. EEG-ITC marker of visual and auditory stimuli were identified in the occipital and temporal lobes of the brain. Our study also suggested that visual stimulus can restore or evoke oscillatory responses in the auditory cortex. Besides, the EEG-ITC coherence increased in the delta and theta bands over the temporal lobe of the brain was recognized as a neural marker of human inhibition. The increased brain connectivity in the frontal lobe of the brain showed the strength of neural connection under human inhibitory control. Finally, an inhibitory brain network model under human response inhibition of left-hand and right-hand using visual and auditory stimuli was proposed, which is supposed helpful for understanding the neural network pathways under inhibition.

## Figures and Tables

**Figure 1 sensors-20-01722-f001:**
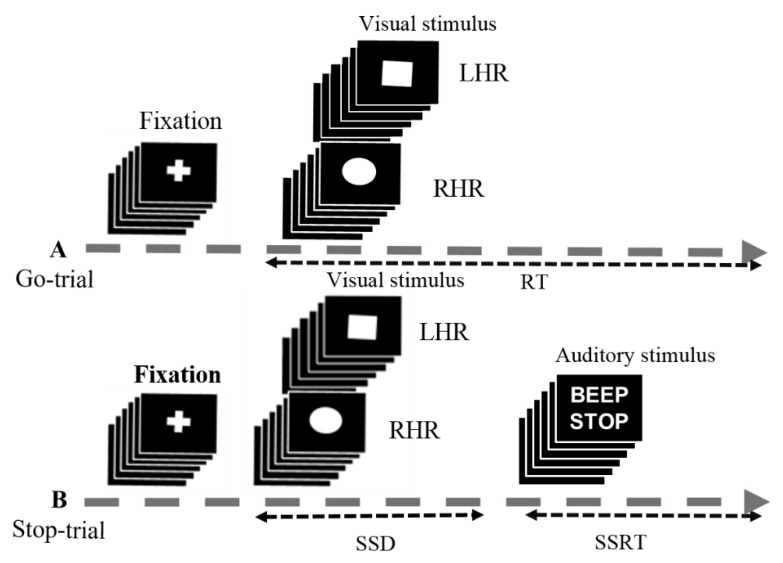
Experimental setup used for stop-signal task. (**A**) In the go-trial (75%), participants respond to the shape of a go stimulus, like a “square” requires the left-hand response (LHR) and a “circle” requires the right-hand response (RHR). The square and circle shapes were used as visual stimuli. (**B**) In the stop-trial (25%), a beep sound (auditory stimulus) was used as a stop signal. Participants were instructed to inhibit the hand response after hearing a beep. In this experiment some parameters were used in the go and stop trials, including the reaction time (RT), stop-signal delay (SSD), and stop-signal reaction time (SSRT).

**Figure 2 sensors-20-01722-f002:**
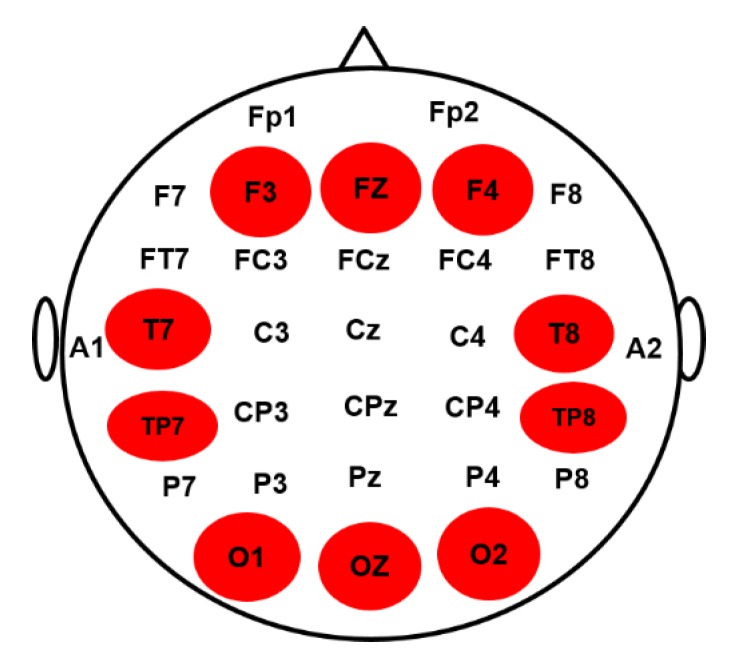
The electroencephalography (EEG) channels placement in a 32-channel EEG cap according to the international 10–20 system were used for data collection. The highlighted brain regions including frontal cortex (F3, FZ, F4), occipital cortex (O1, OZ, O2), left temporal cortex (T7 TP7) and right temporal cortex (T8, TP8) were used for brain connectivity analysis.

**Figure 3 sensors-20-01722-f003:**
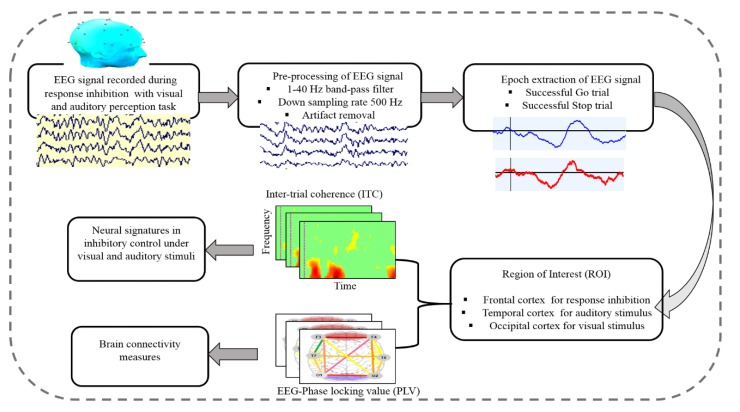
Flowchart of the steps of EEG signal processing for measuring the brain connectivity under human inhibition.

**Figure 4 sensors-20-01722-f004:**
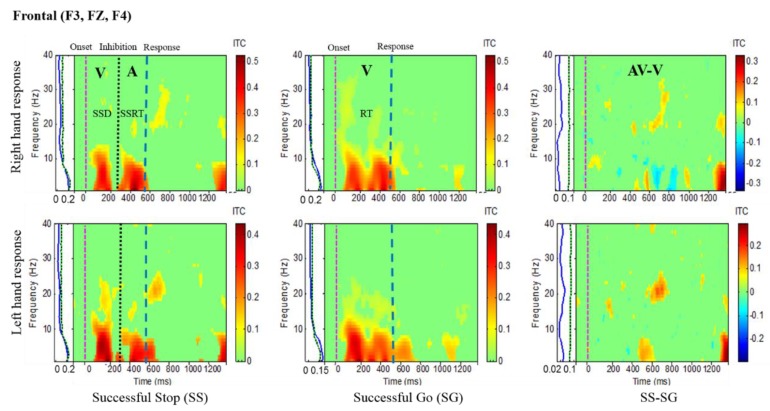
Inter-trial coherence (ITC) plot of frontal cortex (F3, FZ, F4) of the brain with visual (V) and auditory (A) stimuli under left- and right-hand response inhibitions of all male subjects. Purple dashed line: Onset of the go stimulus. Black dashed line: Onset of the stop signal. Blue dashed line: Onset of response. The significant and non-significant ITC values are colored red and green, respectively. The successful go (SG) trial was elicited by only visual stimuli, and the successful stop (SS) trial was elicited by both visual and auditory stimuli. The SS-SG plots display the ITC under human inhibitory control.

**Figure 5 sensors-20-01722-f005:**
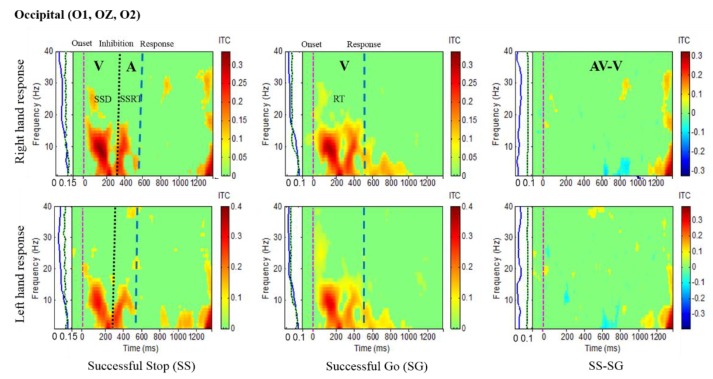
The ITC plot of occipital cortex (O1, OZ, O2) with visual (V) and auditory (A) stimuli under left- and right-hand response inhibitions of all male subjects. Purple dashed line: Onset of the go stimulus. Black dashed line: Onset of the stop signal. Blue dashed line: Onset of response. The significant and non-significant ITC values are colored red and green, respectively. The successful go (SG) trial was elicited by only visual stimuli, and the successful stop (SS) trial was elicited by both visual and auditory stimuli. The SS-SG plots display the ITC under human inhibitory control.

**Figure 6 sensors-20-01722-f006:**
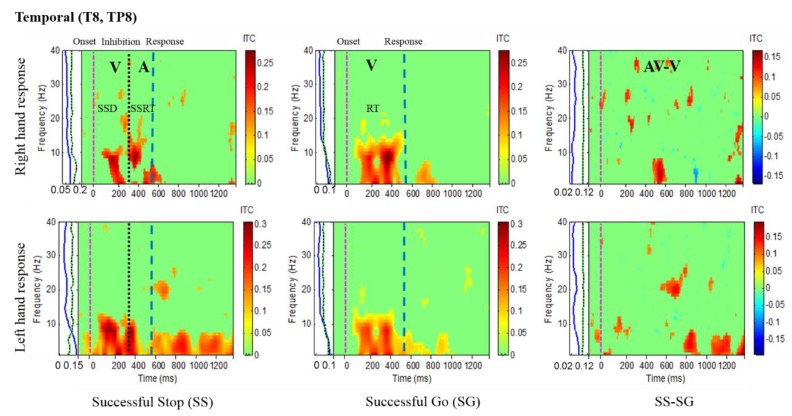
The ITC plot of the right temporal cortex (T8, TP8) of the brain with visual (V) and auditory (A) stimuli under left-and right-hand response inhibitions of all male subjects. Purple dashed line: onset of the go stimulus. Black dashed line: Onset of the stop signal. Blue dashed line: Onset of response. The significant and non-significant ITC values are shown in red and green colors. The successful go (SG) trial was elicited only by visual stimuli, and the successful stop (SS) trial was elicited both by visual and auditory stimuli. The SS-SG plots display the ITC under human inhibitory control.

**Figure 7 sensors-20-01722-f007:**
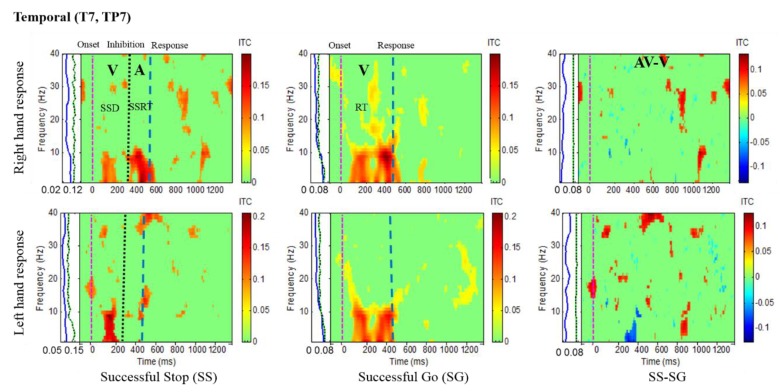
ITC plot of the left temporal cortex (T7, TP7) of the brain with visual (V) and auditory (A) stimuli under left- and right-hand response inhibitions of all male subjects. Purple dashed line: Onset of the go stimulus. Black dashed line: Onset of the stop signal. Blue dashed line: Onset of response. The significant and non-significant ITC values are colored red and green, respectively. The SS-SG plots display the ITC under human inhibitory control.

**Figure 8 sensors-20-01722-f008:**
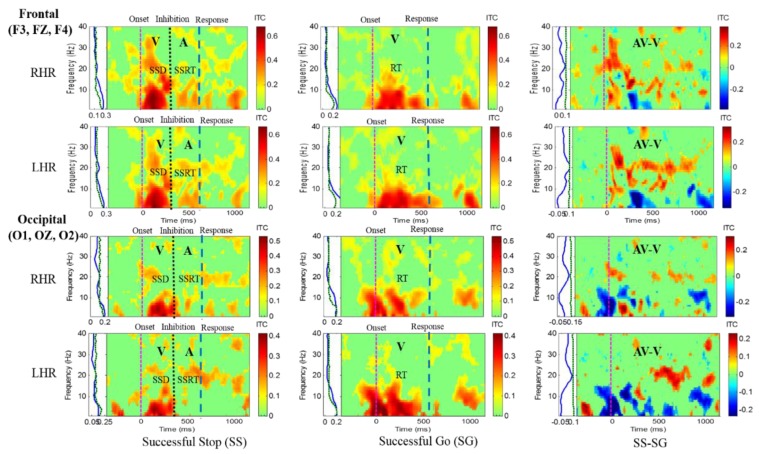
The ITC plot of a single female subject over the frontal cortex (F3, FZ, F4) and occipital cortex (O1, OZ, O2) with visual (V) and auditory (A) stimuli under right hand response (RHR) and left hand response (LHR). Purple dashed line: Onset of the go stimulus. Black dashed line: Onset of the stop signal. Blue dashed line: Onset of response. The significant and non-significant ITC values are shown in red and green colors. The successful go (SG) trial was elicited only by visual stimuli, and the successful stop (SS) trial was elicited both by visual and auditory stimuli. The SS-SG plots display the ITC under human inhibitory control.

**Figure 9 sensors-20-01722-f009:**
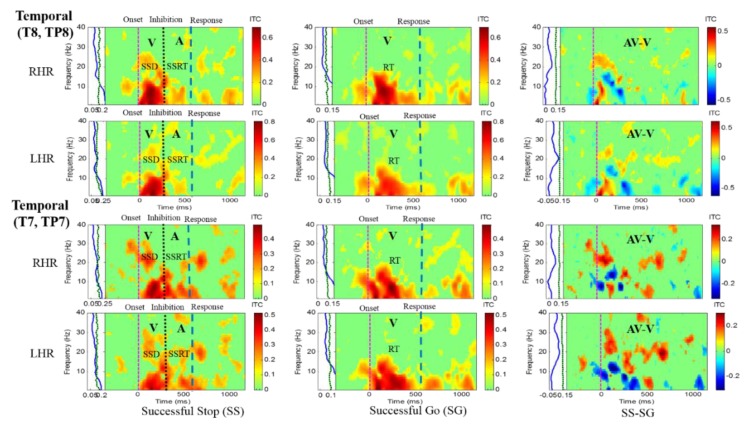
The ITC plot of a single female subject over the right temporal cortex (T8, TP8) and left temporal cortex (T7, TP7) with visual (V) and auditory (A) stimuli under right hand response (RHR) and left hand response (LHR). Purple dashed line: Onset of the go stimulus. Black dashed line: Onset of the stop signal. Blue dashed line: Onset of response. The significant and non-significant ITC values are shown in red and green colors. The successful go (SG) trial was elicited only by visual stimuli, and the successful stop (SS) trial was elicited both by visual and auditory stimuli. The SS-SG plots display the ITC under human inhibitory control.

**Figure 10 sensors-20-01722-f010:**
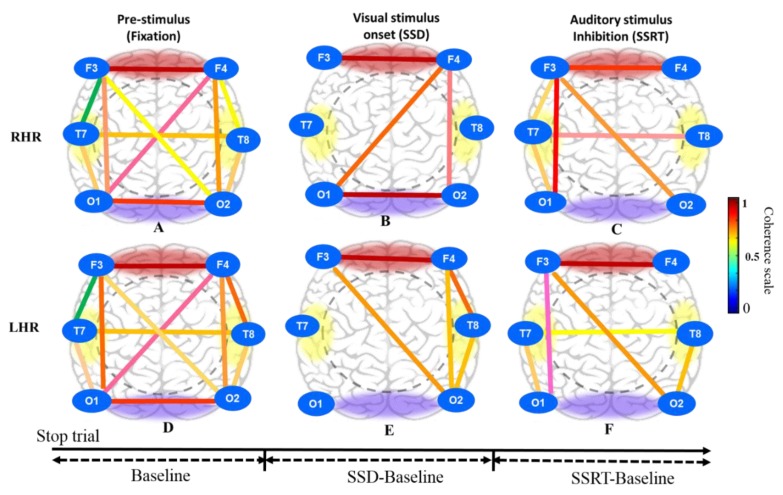
Brain connectivity model in three different conditions inclusive of pre-stimulus (fixation), visual stimulus (SSD) and auditory stimulus (SSRT) in successful stop (SS) condition under right hand response (RHR) and left hand response (LHR) inhibitions of all male subjects. (**A**) Pre-stimulus (fixation); (**B**) Visual stimulus onset (SSD); (**C**) Auditory stimulus (SSRT) during RHR inhibition. (**D**) Pre-stimulus (fixation); (**E**) Visual stimulus onset (SSD); (**F**) Auditory stimulus (SSRT) during LHR inhibition.

**Figure 11 sensors-20-01722-f011:**
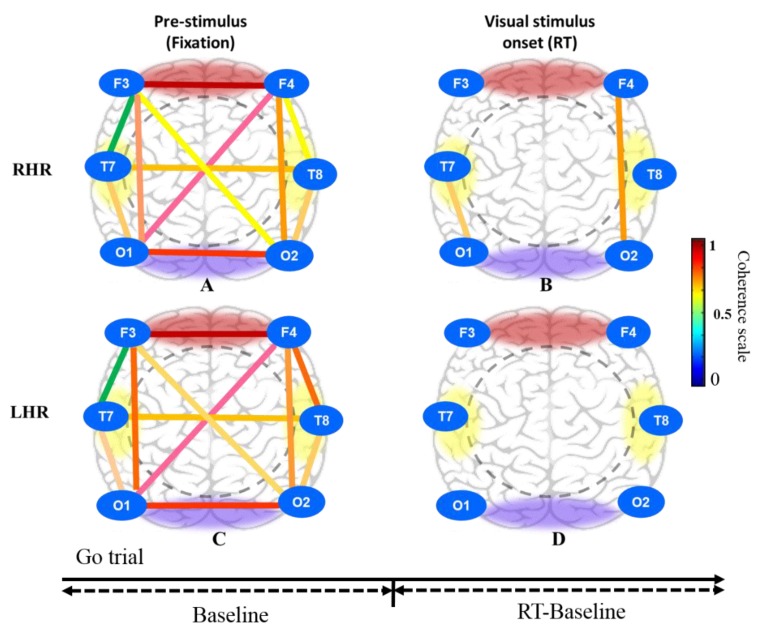
Brain connectivity model in three different conditions that include pre-stimulus (fixation) and visual stimulus in successful go (SG) condition under RHR and LHR inhibition of all male subjects. (**A**) Pre-stimulus (fixation); (**B**) Visual stimulus onset (RT) during RHR inhibition. (**C**) Pre-stimulus (fixation); (**D**) Visual stimulus onset (RT) during LHR inhibition.

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
