# Peer review of "Exploration of Brain Connectivity during Human Inhibitory Control Using Inter-Trial Coherence"

_sensors, 2020, doi:10.3390/s20061722_

Round 1
Reviewer 1 Report
In this study the authors investigated brain connectivity under human inhibitory control using the phase lag index and inter-trial coherence (ITC). The subject of the paper is interesting and hopefully it will be eventually useful.
Some minor revisions are as follows:
- In line 79, all subjects were male. Why no female subjects?
- In line 80, how to know subjects can perform RHR and LHR inhibitions? Please describe the testing process.
- In line 80, “left hand response (RHR)” should be “left hand response (LHR)”.
- In line 155, please describe the parameters of Equation 1, like t, f, N, k, etc.
- In line 185, why were the 11 pairs and 1-40Hz frequency range selected? Please add the explanation or reference.
Author Response
We have revised the all statements of the manuscript carefully, according to reviewer suggestions. The authors are grateful to the editor and reviewer for the constructive comments on this manuscript.

Reviewer 2 Report
The submission Sensors- 730118, entitled “Exploration of Brain Connectivity during Human Inhibitory Control Using Inter-Trial Coherence” by Rupesh Kumar Chikara, and Li-Wei Ko examines brain connectivity under human inhibitory control using the phase lag index and inter-trial coherence. Specifically, using the auditory stop-signal task EEG measurements were generated from 12 male subjects. The results showed that inter-trial coherence in delta and theta bands is increased over the frontal and temporal lobe of the brain. Furthermore, highest brain connectivity was measured under inhibitory control in the frontal lobe compared to temporal and occipital lobes.
According to my opinion, both the topic examined and the experimental results are very interesting. On the other hand, both the research methodology and the presentation quality of this article must be improved.
Specifically, the authors of this article should do the following:
- Although both left hand response and right response were considered, all subjects, which participated in the experimental procedure, were right handed. For this reason, the authors are asked to repeat the same experimental procedure with the participation of some left handed subjects too.
- In many similar case studies the differences between male and female subjects has been investigated. However, all subjects, which participated in the experimental procedure, in this research work, were male. Therefore, the authors are asked to have the same experimental procedure be repeated with female participants.
- The quality of presentation will be enhanced if the experimental results are summarized by using relevant tables.
- The authors must perform thorough editing of English language. Indicatively, the following corrections should be made:
The sentence in lines 64, 65 should be rephrased in order to make sense.
In line 155, the parameters used in equation (1) should be explained, similarly to what happens with equation (2).
The sentence in lines 375-377 should be rephrased in order to make sense.
Wherever more than 2 references are cited, only the numbers of the first and the last of them should be referred
Also,
Line |
Instead of |
Write |
90, 91 |
auditory stop signal task, it was a random selection of go and stop trials. |
auditory stop signal task, which was a random selection of go and stop trials. |
114 |
for each participants. |
for each one of the participants. |
159 |
The ITC method show more accurate results |
The ITC method shows more accurate results |
215,219,222, 229,233 |
increase ITC |
increase of ITC |
221 |
These increase ITC activities |
These increases of ITC activities |
237 |
delta, theta and alpha (8-13Hz) |
delta (1-4Hz), theta (4-8Hz) and alpha (8-13Hz) |
263 |
8A-7F. |
8A-8F. |
266, 282 |
increase neural connectivity |
increase of neural connectivity |
268, 294 |
the increase brain connectivity |
the increase of brain connectivity |
294 |
present |
presents |
311 |
[34], [35], [36], [37]. |
[34-37]. |
316 |
[38], [39], [40], [41]. |
[38-41]. |
348 |
[52-54], [55], [56], [57], [58]. |
[52-58].. |
360 |
adapted from |
adopted from |
Author Response

(The authors gave the same response as above.)

Round 2
Reviewer 2 Report
During the previous round of the reviewing process I emphasized 4 issues that, according to my opinion, should be addressed.
It is very disappointing that the authors, took into consideration only the fourth of them, by simply accepting the numerous grammar corrections that I have highlighted.
Instead, the authors
1) Refused to support their experimental procedure by adding any left handed subject, claiming that it is very difficult to find people of this kind.
2) Refused to support their experimental procedure by using any female handed subject, although it is scientifically well known that the differences in the eeg behavior between the two genders are not ignorable. Their respond, and particularly the 2nd and 3rd sentence of it are somehow difficult to understand.
3) Made no effort to support the comprehension of their experimental results that are illustrated in figures, by further explanation or quantification as requested. Instead, they answered that they have verified and reviewed the presentation of all experimental results by English native speaker.
Author Response
Dear Editor,
Please see the attached file for our responses. We have revised the all statements of the manuscript carefully, according to reviewer suggestions. The authors are grateful to the editor and reviewer for the constructive comments on this manuscript. We believe all of the valuable suggestions have been addressed in our response and in the revised manuscript.

Round 3
Reviewer 2 Report
The article has been enhanced.